# The Effect of Governance Rules Application Controls and the Accrual Basis Application Controls on Quality of Financial Reporting: Applying to Jouf University

**Taha Khairy Taha Ibrahim and Ebrahim Mohammed Al-Matari** \*

Department of Accounting, College of Business, Jouf University, Sakaka 72388, Al-Jouf, Saudi Arabia; tkhtaha@ju.edu.sa
\* Correspondence: emalmatri@ju.edu.sa

**Abstract:** The issue of governance and its applications in Saudi universities is one of the most important issues that has an effective impact on achieving excellence in performance, and achieving societal development for Saudi society. The application of governance rules works to enhance the values of justice, equality, the rule of law, combating corruption, transparency and accountability to contribute to the province on public funds and achieving quality and excellence in performance. So, this study aims to study the impact of the application of governance rules and the accrual basis on the quality of financial reports in Jouf university (Saudi Universities). A total of 348 questionnaires were issued to respondents, with 242 being returned. To put the hypothesized model to the test, structural equation modeling was used. The major findings confirmed the effect of governance rules and accrual foundation of application controls on the quality of Jouf University's financial reporting. The findings have numerous practical ramifications, including the ability to assist managers in making sound decisions when selecting whether to establish governance rules and accrual basis of application controls in their organization. This is a rare and one-of-a-kind empirical study that investigates the impact of governance rules and accrual basis of application controls on the quality of financial reporting at Saudi universities such as Jouf University. This is the first study to give empirical evidence on the relationship between governance rules, the accrual basis of application controls, and financial reporting quality in the context of universities.

**Keywords:** governance rules; accrual basis of application controls; quality of financial reporting; universities



## 1. Introduction

At the local and global level, attention has increased in recent years to the issue of implementing governance rules and the basis of entitlement in universities, as the rules of governance in universities are based on the standards of disclosure, transparency, accountability, independence, and participation of all those involved in managing university affairs in a manner that enables them to develop financial and administrative performance and protect the interests of all parties, and the increased demand for higher education adopted by the Kingdom of Saudi Arabia in recent years has resulted in pressures to be more effective and responsive to community organizations and to achieve the vision and mission of universities in providing high-quality educational output commensurate with the labor market in the KSA in light of the 2030 vision [1].

It should be noted that Saudi universities have played a significant role in establishing a culture of governance in Saudi society by providing an ethical, behavioral, psychological, and value charter for students of higher education, academics, and administrators, and by achieving distinction in financial and administrative performance, which has facilitated the transfer of governance principles and rules to all members of society surrounding the university from the university. This was accomplished through different means such

as seminars, conferences, and articles that promote a disciplined, balanced, just, and controlled culture [2].

The application of governance rules works to enhance the values of justice, equality, rule of law, anti-corruption, transparency, and accountability in order to contribute to preserving public money and achieving quality and excellence in performance. Thus, the Kingdom of Saudi Arabia will adopt the transition to international reporting standards and apply the basis of merit in the government sector which will lead to the improvement of the quality of financial reports, resulting in keeping pace with the rapid economic developments, making bold decisions, and planning for the future by diversifying sources of income and requiring government sectors to make independent final accounts, cash flow statements, and income statements in a way that enhances the principle of disclosure and transparency for all relevant parties by clarifying public expenditures and public revenues. Moreover, it preserves public money from misuse, embezzlement, or loss, as the process of converting to the merit system in Saudi universities is not an easy matter, as it needs to develop many accounting policies to suit the various activities of the university units and so that they do not conflict with internationally recognized accounting principles.

The next part aims to address the problem of study. Moreover, it presents the rules of applying governance in accounting thought and application, followed by addressing the application of accounting on the basis of accrual. The next part devoted the proposed framework to measure the impact of applying the rules of governance and the accrual basis, which contributes to improving financial performance and increasing the quality of financial reports, followed by an analysis of data from the field study and testing of hypotheses. The final part presents the discussion and conclusion.

In recent years, interest has increased in the subject of applying governance rules and the basis of entitlement in universities at the local and global levels, as the rules of governance in universities are based on the adoption of standards of disclosure, transparency, accountability, independence, and the participation of all stakeholders in managing university affairs in a way that enables them to develop financial and administrative performance and protect the interests of all. The increase in demand for higher education that the Kingdom of Saudi Arabia has adopted in recent years has resulted in many pressures to be more effective and responsive to community organizations and to achieve the vision and mission of universities in providing high-quality educational outputs commensurate with the labor market in the Kingdom of Saudi Arabia in light of the Kingdom's Vision 2030. Among the programs of the Kingdom's Vision 2030, the program for strengthening the governance of government work came in recognition of the state's adoption of the governance of its institutions in a manner that ensures the achievement of the required limit of governance rules related to transparency, accountability, participation, flexibility, independence, and disclosure [1].

It should be noted that Saudi universities have a major role in spreading the culture of governance in Saudi society by providing an ethical, behavioral, psychological, and value charter for higher education students, academics, and administrators, and achieving excellence in financial and administrative performance, which resulted in the transfer of the principles and rules of governance to all members of the community surrounding the university from during seminars, conferences, and articles that spread the culture of discipline, balance, justice, control, and follow-up [2].

The application of governance rules works to enhance the values of justice, equality, the rule of law, combating corruption, transparency, and accountability to contribute to preserving public money and achieving quality and excellence in performance to limit profit management practices [3].

There is no doubt that the Kingdom of Saudi Arabia's adoption of the transition to international reporting standards and the application of the accrual basis in the government sector will lead to an improvement in the quality of financial reports, which has resulted in keeping pace with rapid economic developments, taking bold decisions, and planning for the future, by diversifying sources of income and requiring government sectors to

make independent final accounts. Additionally, cash flow and income lists in a way that enhances the principle of disclosure and transparency for all relevant parties by clarifying public expenditures and public revenues and preserving public money from misuse, embezzlement, or loss, as the process of switching to the accrual system in Saudi universities leads to better results, and the application of accounting is considered the accrual basis and is less subject to manipulation compared to the cash basis [4]. Providing accurate and objective information on the accrual basis leads to the achievement of effective and comprehensive control [5], and the process of switching to the accrual basis is not easy, as it requires the development of many policies. Accounting is to be compatible with the various activities of the university units so that it does not conflict with the relevant principles of internationally recognized captivity.

**2. The Rules of Applying Governance in Accounting Thought and Application**

*2.1. The Concept and Importance of Applying Governance*

University governance is one of the modern concepts that has received great attention in recent years due to universities' entering the field of investment and improving the university's own financial resources. The legal and regulatory framework in which all universities operate is through the presence of a governing regulatory body [5]. Some studies [6,7] have identified three guiding principles that help achieve the administrative effectiveness of universities, which are:

- Achieving independence in making decisions and developing future plans in light of a partnership relationship with the government, society, and stakeholders.
- Disclosure, transparency, and speed of response to changes.
- Protecting the academic freedom of educational institutions.

*2.2. Mechanisms and Rules for Applying Governance in Universities from an Accounting Perspective*

2.2.1. Board of Directors (University Council)

The board of directors is the one who manages the affairs of the university based on a delegation. Thus, the final responsibility for the university remains with the board. This means that the board of directors maintains final control over the decisions of the departments, colleges, and supporting deanships, and the results of this study have showed that there is a positive relationship between the number of board members and the manipulation of financial reports; the greater the number of board members, the greater the probability of profit management practices, and consequently the low quality of profits, and the existence of a negative relationship between the proportion of board members from abroad (independent members) and the quality of financial reports [8]. Thus, the characteristics of the board of directors (university council) are summarized as follows:

A. The council demonstrates a commitment to accountability and transparency.
B. The council works to clarify the authority and responsibilities of the university president, his deputies, and the deans of affiliated colleges and centers.
C. The council works on evaluating the performance of the higher leadership positions at the university.
D. The board shall show the results of the final account.

2.2.2. Audit Committees

Audit committees are considered one of the most important mechanisms of governance in universities, as they play a pivotal role in setting accounting policies. The audit committee undertakes a set of tasks. The most important of which are:

- Studying the internal control structure and ensuring its effectiveness on an ongoing basis study.
- Internal audit reports and follows up on the actions taken regarding his recommended observations.

- The board of directors (the university council) has the right to appoint or dismiss the auditor and determine his fees.
- Studying the audit program with the external auditor and making his comments thereon.
- Studying the initial draft of financial statements before submitting them to the board of directors.
- Studying the used accounting policies and expressing opinions and recommendations to the University council.
- Studying the auditor's report on the financial statements and discussing them with its notes [9]. Some studies have found that the availability of financial expertise among the members of the audit committee has a positive relationship with the quality of the financial report [10].

It is worth noting that, the amended regulations for financial affairs in universities have been issued by the decision of the University Affairs Council No. (1-4-1442) dated in 9/14/1442 AH, corresponding to 04/26/2021 AD, and the twenty-eighth article regarding financial control after disbursement, paragraph (1), which stipulates that a decision of the university council constitutes an audit committee that emanates from it and is subordinate to it with the purpose of following up the work of the external auditors and the supervision of the internal audit working at the university in accordance with the provisions of the six regulations specified as the tasks of the committee.

### 2.2.3. Internal Control and Audit

An internal audit is an independent evaluative activity that examines and evaluates the accounting and financial aspects of the facility for the purpose of serving the management, maintaining the project's assets, determining the accuracy of the accounting data, and detecting violations and abuses in government units, with noticing that, the Council of Ministers Resolution No. 129 of 1428 AH approved the regulation. Each entity undertakes the establishment of an internal audit unit at the main headquarters, which is directly linked to the responsible manager in the governmental unit.

### 2.2.4. Auditor of Accounts

The main function of the external auditor is to achieve oversight over the financial reports of the government units with the aim of enhancing the credibility of the accounting information presented in the financial statements by ensuring that, it is accurate, free from bias, and that it reflects the reality of financial conditions. Therefore, the auditor is considered one of the mechanisms of the governance, and supports its effectiveness by reducing potential risks. There is a relationship between the auditor and audit committees in the government units in accordance with the financial regulations governing university affairs. The audit committees are responsible for appointing him, determining his fees, coordinating with him, following up on his work, studying his observations and views on the final account—taking them into account—and informing relevant authorities. The relationship between the implemented observations and their remediation in the future and the submission of recommendations in this regard should be introduced to the university council (Regulations regulating financial affairs in universities, No. (1/4/1442), 2021.

### 2.2.5. Stakeholders

Stakeholders represent internal and external parties affected by the decisions of the university council. The standards for implementing the governance stipulated by the Organization of Economic Cooperation and Development included the role of stakeholders. They clarified the need to work to respect their legal rights as well as the mechanisms for their effective participation in providing job opportunities and monitoring units (Organization of Economic Cooperation and Development, 2006).

The role of stakeholders or related parties was also mentioned in the International Principles for Corporate Governance issued in April 2004, which clarified that the framework of the rules governing the governance must recognize the rights of stakeholders as

defined in the law, and encourage the effective cooperation between government units and stakeholders to create jobs and ensure the continuity of the government boards financially.

Accordingly, the researchers see the commitment of the board of directors (university council) must adhere to transparency and disclosure of financial and non-financial information and the participation of all stakeholders in preparing the final account and defining educational, administrative, and financial policies. This leads to the stakeholders' reassurance of the possibility of relying on financial reports prepared by the administration, and the credibility of the work. Additionally, it helps to improve financial performance.

## 3. The Rules of Applying the Accrual Basis in Government Units

### 3.1. Accounting Measurement for Accrual Basis

The accrual basis means to prove the expenses pertaining to the accounting period, whether they have been paid or not, and also the revenues pertaining to the accounting period, whether they have been received or not [11]. Contrary to the cash basis which recognizes expenses as soon as they are paid to others, whether those expenses are pertained to the current period or not. Recognizing revenue occurs when its value is received and collected from others, whether it is pertained to the current period or not. Thus, the monetary basis does not show an adequate picture of the actual reality of all financial operations which results in showing inaccurate results on the revenues, expenses, and expenditures of government units. It does not show full information about the resources required to finance projects and services. Accordingly, we review the most important advantages of applying the accrual basis as follows:

- Provide an accurate picture of the financial situation of government units.
- Financial performance appraisal.
- Preparing financial reports.

### 3.2. The Desired Objectives of Applying the Accrual Basis

- Enhancing the principle of transparency, accountability, and decision-making support by providing more accurate and comprehensive information to enhance planning processes and access to accurate and complete financial information on the costs of government services and their future obligations during achieving oversight and sharing information with the community.
- Preparing an inventory of all assets and liabilities of government agencies, and showing the financial position of the agencies independently.
- This shows the final account of the universities independently and helps in the possibility of comparing the financial performance, comparing the budget with the actual numbers of the public sector and comparing the actual numbers of the public sector with previous years.
- It helps in setting accounting policies and thus measuring performance for successive years according to unified accounting principles, as well as linking the achieved performance and the cost spent on it.
- Enabling the university council and stakeholders to compare the results of financial performance and take appropriate decisions.

## 4. Quality of the Financial Reports

The current study aims to study the impact of applying the rules of governance and the basis of accrual on the quality of financial reports in Saudi universities (Jouf University).

Therefore, we are going to define the quality of financial reports and their characteristics to complete the requirements of the study according to the vision of the International Accounting Standards Board in the public sector. Through the survey of researchers, it has been found that the university will apply the accrual basis in accordance with the international accounting standards in the public sector. Qualitative characteristics of financial reports in the public sector for general purposes (IPSASBs, [12]).

Qualitative qualities are the traits that make the information shown in financial statements relevant to users, and they apply to the statements independently of the accounting foundation used to create the financial statements. They are as follows:

- Appropriateness;
- Reliability;
- Neutrality;
- Prudence and Caution;
- Completeness;
- Understandability;
- Comparability.

## 5. Literature Review and Methodology

This section highlights the main theory that explains the association between independent variables and dependent variables. the theory is called agency theory. The foundations of agency theory are a set of assumptions and fundamentals that reflect the agency relationship between the two parties, the agent and the principle. The rule is also based on the existence of a contractual relationship between two main parties, the agent who serves the principal party and the principal party who authorizes the agent to perform on his behalf, to perform his duties, in accordance with the principal party's best preference and interest [13].

Companies use governance as one of the mechanisms related to agency theory to ensure that the behavior of executive managers and workers within companies is controlled, in order to achieve the interests of owners and shareholders while avoiding special interests [14]. Furthermore, the government is based on incentive rules, which motivate agents to act in the interests of the entrusting party rather than their own personal interests, as well as the existence of rules to control behavior, which achieve separation between the agent's interests and the interests of the clients, in a manner that achieves the company's basic interest, which is to make a profit, and the interests of the shareholders in this company [15]. Finally, this next part aims to deal with previous studies related to the current study as follows.

### 5.1. Research on Governance Mechanisms

**Carcello, Hollingsworth, Klein, and Neal** [16] aimed to explain and analyze the relationship between the financial audit experience and the mutual mechanisms of corporate governance and profit management. Additionally, it concluded that the different approaches to corporate governance help manage the effectiveness of the quality of financial reports. The study recommended the need to expedite the use of different approaches to corporate governance to improve the quality of financial reports.

**Goodwin-Stewart and Kent** [17] aimed to explore the voluntary use of internal auditing in Australian companies listed on the stock exchange that apply corporate governance and concluded that there is a strong relationship between internal auditing and the level of commitment to risk management. It also confirmed that the limitations of the use of internal audit by Australian companies had important reflections on corporate governance for those that use internal audit.

**Kassab and Al-Razeen** [18] aimed to determine the role of governance mechanisms, whether internal (as stated in Saudi Corporate Governance Regulations issued in 2006) or external (with financial institutions within the company's ownership structure) in enhancing the quality of financial reports. A sample of Saudi joint stock companies was selected to conduct the test for the years 2005, 2006, 2007, and 2008. The quality of profits was measured before and after the issuance of the governance regulation, and the results also confirmed that the presence of financial institutions within the structure of the company's ownership structure leads to an increase in profits and then the quality of the financial reports. The study recommended the necessity of adhering to the governance regulation and not continuing to be a guiding regulation only.

**Sami** [19] sought to investigate and assess the role of audit committees in improving corporate governance effectiveness and its impact on the quality of financial statements presented in the Egyptian business environment. It urged that the obligations of the members of the board of directors are supported in order to meet the goals of the corporate governance framework.

**Hammad** [2] aimed to show the role of universities in spreading the culture of governance in society by presenting the concepts and characteristics of governance and the experiences of developed countries to activate corporate governance. It focused on the importance of the role of universities in spreading the culture of governance in civil society through students, graduates, faculty members, etc., paying attention to moral and psychological structure where satisfaction, loyalty, excellence, and innovative thought are attained. It recommended the necessity of holding brainstorming sessions for all university employees to present ideas and proposals that discuss practical problems in society, and the need for monitoring and follow-up systems from all parties associated with the organization to reduce corruption.

**Giovanna** [20] aimed to highlight a model of university governance after applying the law on the formation of the board of directors in Italian universities, as it required new competencies and good practices in the field of management. It recommended the necessity of amending the organizational structure of universities in accordance with the new law to keep pace with the rapid developments.

**Akari and Bousselma** [21] sought to demonstrate the impact of corporate governance on the quality of accounting information, and concluded the most important reason for implementing corporate governance is to restore confidence in accounting information and tighten control over it by achieving accountability and control, and striving towards the development and application of accounting standards.

**Al-Qudah, Kharabsheh and Hamidat** [22] aimed to show the impact of corporate governance on the degree of caution and caution in the financial statements published in Jordanian commercial banks. There are no statistically significant differences in the degree of prudence and caution for profits. It recommended that the supervising authorities of Jordanian commercial banks work with the concept of caution and caution in accordance with Jordanian laws and legislation, and the requirements of international financial reporting standards in measuring levels of caution and caution in the Jordanian business environment.

**Al-Suwaidi** [23] aimed to determine the impact of corporate governance on the level of disclosure in accounting information. The Amman Stock Exchange was chosen to conduct this study, disclosing them and publishing standards of ethical behavior for service companies.

**Omran and Abu Al-Walafa** [24] aimed to identify whether the international financial reporting standards lead to the improvement of the corporate governance process through the role of standards in supporting the basic pillars of governance, which are the management of risks to which the organization is exposed, and the provision of an appropriate environment for monitoring the performance of the administration; disclosure, transparency, and management motivation. It concluded that international financial reporting standards are the most appropriate to achieve effective governance, and recommended the need to create the necessary climate for the application of international reporting standards with regard to laws and legislation (the development of internal administrative and accounting systems for companies, or the development of scientific and professional qualifications for accountants).

**Abdel Ghaffar** [25] aimed to analyze the mutual impact between financial control in NGOs and governance mechanisms to prevent and reduce corruption. It exposed a relationship between the strength and weakness of the elements of financial control in non-profit organizations and internal and external governance mechanisms. It recommended that non-profit organizations are obligated by disclosing the sources of foreign and domestic funding and expenditures within the financial statements and reports.

**Al-Washah and Shaheen** [26] aimed to measure the impact of applying the rules of governance on accounting disclosure and the quality of financial reports and transparency, collectively and individually at the level of accounting disclosure and the quality of financial reports in private universities. It recommended the necessity of applying the rules of governance in a way that contributes significantly to enhancing the role of accounting disclosure and the quality of financial reports and is positively reflected on the performance of these universities.

**Al-Shamry** [1] aimed to identify the governance of universities and the application of their indicators and their role in achieving the Kingdom's Vision 2030. The study used a descriptive approach that relied on the analysis of the higher education system and what it included in terms of regulations, programs, and goals. The study concluded that there are many obstacles facing the application of governance indicators in Saudi universities and there is a trend from the government in implementing governance. It recommended the need to pay attention to indicators of governance in higher education institutions.

**Al-Mafeez** [27] aimed to develop a proposed vision for applying governance in Saudi universities to keep pace with developments and give the best scientific and research outputs to the community. The study recommended the necessity of granting universities financial and administrative independence to achieve their vision and mission, promoting the culture of transparency and disclosure, and ensuring the rights of stakeholders or internal and external beneficiaries with the presence of an ethical and professional charter that governs collective behavior.

*5.2. Research into Accrual Accounting*

**Obaidat, Al-Dahyan, and Owais** [28] aimed to identify the impact of applying the accrual basis on the quality of financial reports in institutions subject to the government unit budget law; the accrual basis in government units. It concluded that the most important of which is the effect of applying the accrual basis in Jordanian government sector on the quality of financial reports.

**Jassi** [29] aimed to develop a proposed model for applying the accrual basis in non-profit government units and used the applied study at the University of Al-Muthanna in Iraq, to conclude that the application of the cash basis does not achieve the objectives of government accounting represented in planning, decision-making, performance evaluation, and achieving effective control over the disbursement of expenses and collection of revenues.

**de Sousa et al.** [30] aimed to define the concepts of users and preparers of accounting information in the Brazilian public sector about the adoption of the accrual basis and the production of a new system to produce valuable information for decision-makers and managers of public bodies. It concluded that there is a need to increase transparency and accountability in the public sector, and recommended the need to fully rely on the accrual basis which leads to better information for making decisions and evaluating the performance of managers.

**Wynne** [31] aimed to clarify the advantages of moving to the accrual basis from the cash basis due to the presence of many advantages, concluding that the application of the accrual basis helped in knowing the financial position of the government and its ability to finance its activities and fulfill its obligations. The study recommended that the risks of the transition to the accrual basis should be managed in an appropriate way.

**Hassan** [4] sought to reconsider the debates surrounding the accrual accounting in the public sector, and to demonstrate how opinions differ between practitioners and academics. The study discovered that a large percentage of practitioners prefer the accrual accounting while most academics are opposed to it. The study recommended the need to bridge the gap and increase communication and coordination between practitioners and academics.

**Meligy** [32] aimed to test the impact of the transition to international financial reporting standards on the quality of accounting information which depends on the integration of each of the qualitative characteristics such as relevance, reliability, comprehension, and comparability, and quantitative characteristics that focus on the volume of information

provided to investors and the parties benefiting from it. It concluded that there is a positive impact of international financial reporting standards on the quality of accounting information, and recommended the need to complete the creation of a Saudi business environment to complete the transition to financial reporting standards in all economic sectors.

*5.3. Evaluation of Studies and Identification of the Research Gap for the Study*

As for the studies that dealt with the principles and rules governing the application of governance:

1. Each of the previous studies dealt with one of the accounting aspects of applying corporate governance rules, such as the different approaches to corporate governance, the use of internal and external audits, audit committees, governance mechanisms in joint stock companies, and disclosure and transparency policies.
2. Researchers' opinions differed regarding the importance of applying governance and the basis for its recognition to be measured and disclosed in financial reports.
3. Most of the studies dealt with the mechanisms of governance for the private sector and joint stock companies and did not address it in Saudi universities, except [27], which dealt with the mechanisms of governance in universities from an administrative perspective, but not from an accounting perspective.
4. Some studies focused on disclosure and transparency to activate the role of governance and its impact on financial reports neglecting other aspects.
5. Therefore, none of these studies provided an integrated model for the rules of applying governance in Saudi universities.

As for the studies that dealt with accounting on an accrual basis:

1. Some studies have shown the effect of applying the accrual basis on the quality of financial reports in Jordanian government sector. They have indicated the need to expand the use of accrual accounting in the government sector.
2. Some studies focused on defining concepts for the accrual basis, while others focused on clarifying the advantages of moving from the cash basis to the accrual basis because it has many advantages, including the application of transparency and accountability in the government sector. Therefore, none of these studies provided an integrated model for the application of accounting to the accrual basis, and there is no definitive link between the effect of applying the rules of governance and the application of the accrual basis in Saudi universities on the quality of financial reports.

The researchers believe that none of these studies dealt with measuring the impact of applying the rules of governance and the accrual basis on the quality of financial reports, which will be discussed in the current study through the application of a sample of workers in the administrative and financial units and academics who practice this at Jouf University and its impact on the quality of financial reports.

The subject of the current research is one of the new topics that the Kingdom has adopted for its application in the government sector by shifting to the basis of accrual, which helps to evaluate the financial performance of Saudi universities accurately and comprehensively.

The following hypothesis is proposed based on the preceding discussion:

**H1**. There is an influence that is both good and significant of governance rules application controls and quality of financial reporting.

**H1a**. There is a positive and significant effect of governance rules application controls (**board of directors**), and quality of financial reporting.

**H1b**. Governance rules and application controls (audit committees) have a positive and significant impact on financial reporting quality.

**H1c**. There is a positive and significant effect of governance rules application controls (**risk management**), and quality of financial reporting.

**H1d**. Governance rules and application controls (stakeholders) have a positive and significant impact on financial reporting quality.

**H2**. Accrual basis application controls have a positive and significant impact on financial reporting quality.

## 6. Methodology

The hypothesized model was investigated using a survey questionnaire research design and quantitative methods technique. In total, 348 questionnaires were dropped off at the Jouf University to the intended respondents. Jouf University is a company with over 3599 employees (academic, administrative and financial). In addition, the data were analyzed using SPSS software and partial least square structural equation modeling (PLS-SEM).

### 6.1. Measure

The approaches for measuring the variables were drawn from previous literature. Quality of financial reporting measures were developed by [32,33]. Many studies measured accrual basis application controls by [4,29–32]. Moreover, the questions on governance rules application controls were adapted from [1,9,26,27]. A list of the objects used in this investigation may be found in the Appendix A. The responses were measured using a five-point Likert scale ranging from "1" (strongly disagree) to "5" (strongly agree) (strongly agree).

### 6.2. Procedures for Data Gathering and Sampling Design

At Jouf University, 348 personnel (academic, administrative, and financial) were given a drop-off questionnaire. A bilingual person translated the questionnaire from its original language into Arabic. The Arabic version was re-translated into English by another bilingual person to detect any adjustments and changes, as advised by [34] by comparing the two English versions and thus ensuring the validity and reliability of the instrument. Proportionate stratified random sampling was used as the sampling method. Because of the nature and hierarchy of Jouf University, which has many general departments and police stations, this method was chosen. This method generates a sample that is highly representative of the population under investigation, allowing the researcher to extrapolate the findings to the entire population. The Randomizer program, which is available online, and Microsoft Office Excel 2013 were used to select random samples. Moreover, [34] suggested doing a power analysis test to estimate the minimum sample size (Figure 1).

Furthermore, using the G*Power 3.1.9.4 program, an a priori power analysis was performed [35]. Based on the following statistical parameters [36]: a medium effect size f2 (0.15), an alpha significance level (err prob, 0.05), power (1 err prob; 0.95), two predictors (i.e., governance rules application controls, accrual basis application controls), and three main numbers of predictors as a total (i.e., governance rules application controls, accrual basis application controls, and quality of financial reporting (Figure 2).

### 6.3. Research Framework

As indicated in Figure 3, the framework includes two exogenous factors (governance rules application controls and accrual basis application controls) as well as indigenous variables (quality of financial reporting). In this framework, there are two connections between the independent and dependent variables: governance rules application controls and financial reporting quality, and accrual basis application controls and financial reporting quality. These relationships were expected to evaluate the study's conceptual framework, as illustrated above.

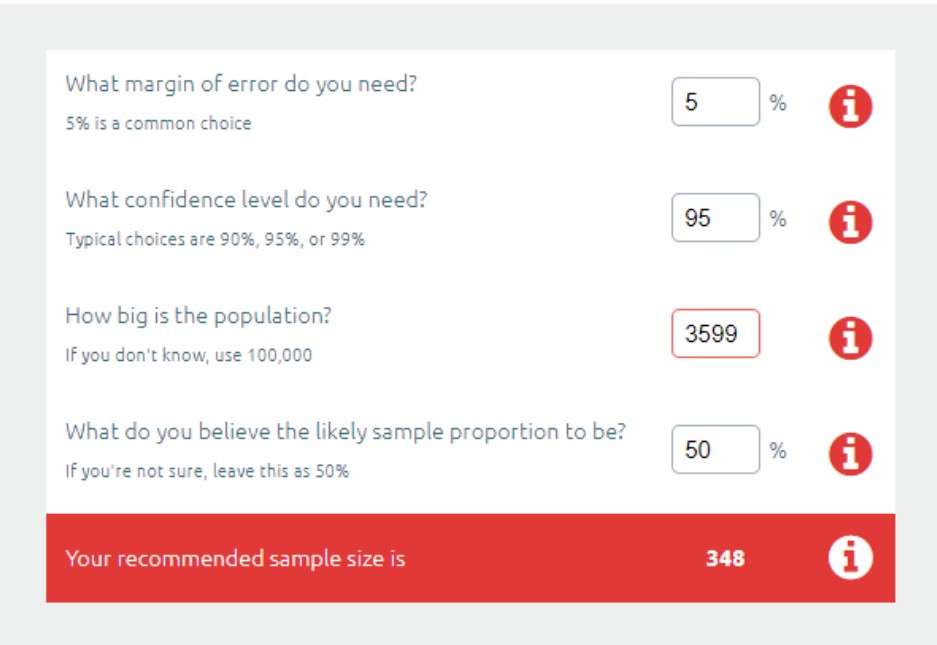

**Figure 1.** How to choose data.

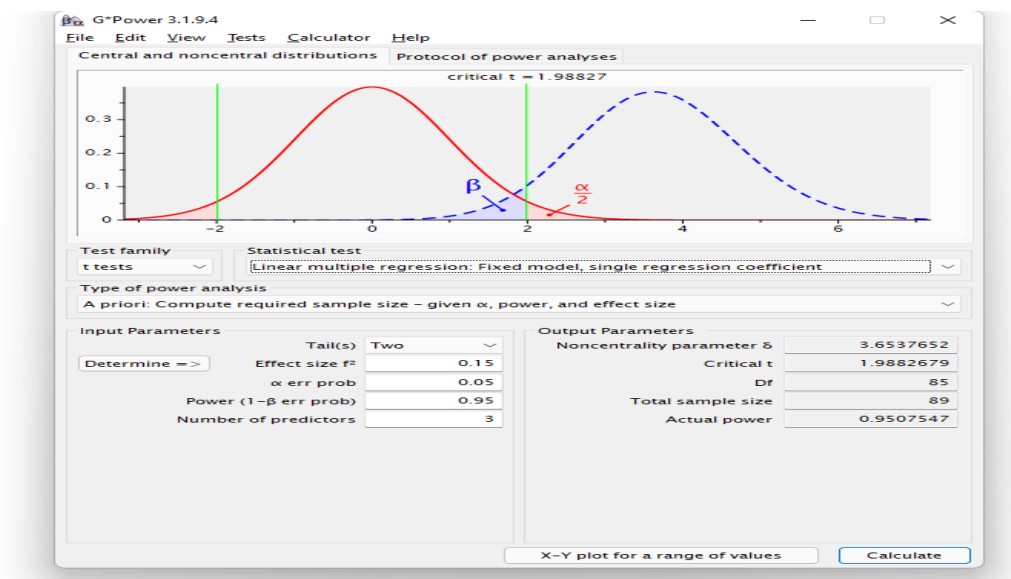

**Figure 2.** How to choose data.

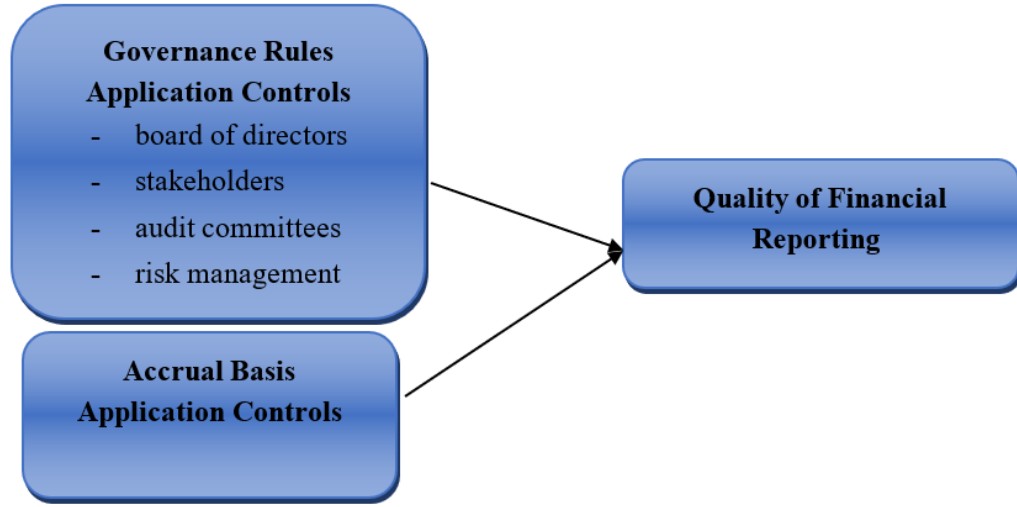

**Figure 3.** Research Framework.

## 7. Data Analysis and Findings

### 7.1. Profile of Respondents

The demographic information of the respondents was divided into four categories, as indicated in Table 1. A total of 242 responses out of 348 were used to compile the data. A 70 percent response rate was recorded in the data collected. As can be seen in the diagram below, there are 218 males (90.1%) and 24 females (9.9%). In terms of qualifications, 56.2 percent of respondents have a PhD, followed by 35.1 percent with a bachelor's degree, and 8.7 percent with a master's degree. The experience demographic is divided into four categories, as illustrated in the table below. Academic personnel account for 60.7 percent of responses, administrative staff for 25.3 percent, and finance employees for 14 percent.

**Table 1.** Participants' Demographic Information.

| Demographic Variable | Category | Frequency (*n* = 242) | Percent % |
|---|---|---|---|
| Gender | Male | 218 | 90.1 |
| | Female | 24 | 9.9 |
| Qualifications | Bachelor | 85 | 35.1 |
| | Master | 21 | 8.7 |
| | PhD | 136 | 56.2 |
| Experiences | 10–less than 20 Years | 61 | 25.2 |
| | More than 20 years | 11 | 4.5 |
| | 5–less than 10 | 112 | 46.3 |
| | Less than 5 years | 58 | 24 |
| Position | Academic | 147 | 60.7 |
| | Administrative | 61 | 25.3 |
| | Finance | 34 | 14 |

### 7.2. Descriptive Analysis

As shown in Table 2 below, descriptive analysis was presented for the construct. The data show that the board of directors has the highest mean, with 4.159, and the lowest standard deviation, with 0.840. The result indicated an agreement among the respondents about the importance of the board of directors and their role in qualifying the financial reporting. However, on the other hand, stakeholders have the lowest mean of 4.075 and the highest standard deviation of 0.945.

**Table 2.** Descriptive Statistics.

| Construct | n | Minimum | Maximum | Mean | Std. Deviation |
|---|---|---|---|---|---|
| Board of Directors | 242 | 1 | 5 | 4.159 | 0.840 |
| Stakeholders | 242 | 1 | 5 | 4.075 | 0.945 |
| Audit Committees | 242 | 1 | 5 | 4.110 | 0.886 |
| Risk Management | 242 | 1 | 5 | 4.084 | 0.861 |
| Accrual Basis Application Controls | 242 | 1 | 5 | 4.143 | 0.882 |
| Quality of Financial Reporting | 242 | 1 | 5 | 4.130 | 0.872 |

*7.3. Structure Equation Modelling Results*

In the literature, partial least square structural equation modeling (PLS-SEM) is widely used as a non-parametric model testing technique.

PLS-SEM was proposed by [37] as a standard method used in path models to estimate causal links and calculate latent components. However, the PLS-SEM approach is essentially a regression sequence for obtaining convergent fixed-point equations. Even though the distribution of test path models is substantially skewed, PLS can predict a restricted number of them [38].

Before assessing hypotheses in the structural model, the measurement model was assessed in this study utilizing model validity and reliability, as mentioned in the following sections.

### 7.3.1. The Measurement, Outer Model

Prior to evaluating the given hypotheses, the estimation model was validated using the PLS-SEM methodology. To that purpose, [39] proposed a two-stage technique, which was used in conjunction with this analysis.

Content validity, convergent validity, and discriminant validity were used to examine the construct's reliability and validity.

### 7.3.2. Content Validity

Content validity is defined in the multivariate analysis literature as when items used to measure a construct display in the same model have higher loads on their constructions than the other constructs. As a result, the loading variable has been employed to verify content validity, as indicated by [40,41]. If goods have dimensions that are greater than their loads, they will be excluded.

All objects loaded more on their respective constructs than the constructs of other forms, according to the data reported in Table 3. The results demonstrate the importance of all the variable products' factor loading on their respective constructs. This result validated the measurement method's content validity.

### 7.3.3. Convergent Validity

The degree to which a group of items converge to measure a specific construct is known as convergent validity [40]. The average variance extracted (AVE), factor loading, and composite dependability can all be investigated in the SEM literature.

As a result, the loading should be strongly loaded and statistically significant when computing constructs with at least 0.7 for variable loading and composite reliability and at least 0.5 AVE, as indicated in Table 4. The results confirmed the model's convergent validity by showing that the results met the cut-off values [38].

Furthermore, construct reliability was assessed by comparing Cronbach's alpha values to composite reliability values, as shown in Table 2. Previous studies, such as [42] and [40] suggested a cut-off value of 0.7. The Cronbach's alpha and composite reliability ratings were both better than 0.7, indicating that the items were adequate for measuring their respective constructs and were reliable.

### 7.3.4. Discriminant Validity

The degree to which a set of items can identify a construct from other constructs in the model is referred to as discriminant validity in the SEM literature. According to [43], the diagonal elements should have larger values (AVE's square roots) than the relevant rows and columns, as seen in Table 5, which validated the discriminant validity and hence the adequacy of measurement.

**Table 3.** Significant Factor Loadings.

| Construct | Item | Loadings | Standard Error | t Value | p Value |
|---|---|---|---|---|---|
| Accrual Basis Application Controls | ABAC1 | 0.927 | 0.018 | 52.749 | 0.000 |
| | ABAC2 | 0.937 | 0.014 | 69.059 | 0.000 |
| | ABAC3 | 0.935 | 0.013 | 73.747 | 0.000 |
| | ABAC4 | 0.950 | 0.010 | 97.920 | 0.000 |
| | ABAC5 | 0.944 | 0.010 | 92.944 | 0.000 |
| | ABAC6 | 0.941 | 0.011 | 85.010 | 0.000 |
| Audit Committees | AC1 | 0.942 | 0.012 | 77.694 | 0.000 |
| | AC2 | 0.968 | 0.006 | 153.826 | 0.000 |
| | AC3 | 0.945 | 0.010 | 91.689 | 0.000 |
| Board of Directors | BD1 | 0.880 | 0.021 | 42.324 | 0.000 |
| | BD2 | 0.911 | 0.019 | 48.654 | 0.000 |
| | BD3 | 0.922 | 0.014 | 64.948 | 0.000 |
| | BD4 | 0.809 | 0.038 | 21.265 | 0.000 |
| Risk Management | RISK1 | 0.932 | 0.012 | 77.364 | 0.000 |
| | RISK2 | 0.944 | 0.011 | 86.740 | 0.000 |
| | RISK3 | 0.926 | 0.026 | 36.073 | 0.000 |
| | RISK4 | 0.895 | 0.030 | 29.818 | 0.000 |
| Stakeholders | STAK1 | 0.876 | 0.027 | 32.998 | 0.000 |
| | STAK2 | 0.911 | 0.017 | 52.742 | 0.000 |
| | STAK3 | 0.881 | 0.019 | 47.066 | 0.000 |
| | STAK4 | 0.824 | 0.028 | 29.241 | 0.000 |
| Quality of Financial Reporting | QFR1 | 0.917 | 0.014 | 66.598 | 0.000 |
| | QFR2 | 0.944 | 0.010 | 96.639 | 0.000 |
| | QFR3 | 0.933 | 0.017 | 55.318 | 0.000 |
| | QFR4 | 0.939 | 0.013 | 74.809 | 0.000 |
| | QFR5 | 0.831 | 0.052 | 15.903 | 0.000 |
| | QFR6 | 0.859 | 0.032 | 26.759 | 0.000 |

**Table 4.** Convergent Validity Analysis.

| Construct | Item | Loadings | Cronbach's Alpha | CR [a] | AVE [b] |
|---|---|---|---|---|---|
| Accrual Basis Application Controls | ABAC1 | 0.927 | | | |
| | ABAC2 | 0.937 | | | |
| | ABAC3 | 0.935 | 0.973 | 0.978 | 0.882 |
| | ABAC4 | 0.95 | | | |
| | ABAC5 | 0.944 | | | |
| | ABAC6 | 0.941 | | | |
| Audit Committees | AC1 | 0.942 | | | |
| | AC2 | 0.968 | 0.948 | 0.966 | 0.905 |
| | AC3 | 0.945 | | | |
| Board of Directors | BD1 | 0.88 | | | |
| | BD2 | 0.911 | 0.903 | 0.933 | 0.777 |
| | BD3 | 0.922 | | | |
| | BD4 | 0.809 | | | |

**Table 4.** *Cont.*

| Construct | Item | Loadings | Cronbach's Alpha | CR [a] | AVE [b] |
|---|---|---|---|---|---|
| Risk Management | RISK1 | 0.932 | 0.943 | 0.959 | 0.854 |
| | RISK2 | 0.944 | | | |
| | RISK3 | 0.926 | | | |
| | RISK4 | 0.895 | | | |
| Stakeholders | STAK1 | 0.876 | 0.896 | 0.928 | 0.763 |
| | STAK2 | 0.911 | | | |
| | STAK3 | 0.881 | | | |
| | STAK4 | 0.824 | | | |
| Quality of Financial Reporting | QFR1 | 0.917 | 0.955 | 0.964 | 0.819 |
| | QFR2 | 0.944 | | | |
| | QFR3 | 0.933 | | | |
| | QFR4 | 0.939 | | | |
| | QFR5 | 0.831 | | | |
| | QFR6 | 0.859 | | | |

[a]: CR = (Σ factor loading) 2/{(Σ factor loading) 2) + Σ (variance of error)}; [b]: AVE = Σ (factor loading) 2/(Σ (factor loading) 2 + Σ (variance of error)}.

**Table 5.** Correlations of Discriminant Validity.

| Construct | Accrual Basis Application Controls | Audit Committees | Board of Directors | Governance Rules Application Controls | Quality of Financial Reporting | Risk Management | Stakeholders |
|---|---|---|---|---|---|---|---|
| Accrual Basis Application Controls | 0.939 | | | | | | |
| Audit Committees | 0.745 | 0.952 | | | | | |
| Board of Directors | 0.742 | 0.711 | 0.882 | | | | |
| Governance Rules Application Controls | 0.841 | 0.903 | 0.903 | 0.830 | | | |
| Quality of Financial Reporting | 0.902 | 0.757 | 0.738 | 0.837 | 0.905 | | |
| Risk Management | 0.833 | 0.854 | 0.753 | 0.932 | 0.801 | 0.924 | |
| Stakeholders | 0.753 | 0.770 | 0.846 | 0.930 | 0.771 | 0.793 | 0.874 |

*7.4. Structural Model (Inner Model) and Hypotheses Testing*

After establishing the measurement model by analyzing different validities in the previous sections, the suggested hypotheses were investigated as structural models by running the SmartPLS algorithm and bootstrapping. The path coefficient and their significance were extracted to confirm the model's adequacy and see whether the hypotheses were confirmed or not.

Figures 4–6 and Table 6 show that governance rules application has a insignificant impact on the quality of financial reporting at 0.01 ($\beta$ = 0.269, t = 4.418, $p < 0.01$); therefore, H1 is supported. The other sub-hypotheses (H1a–H1s) of dimension related to governance rules application, all (board of directors, stakeholders, audit committees and risk management) have a positive significant effect on the quality of financial reporting relationship ($\beta$ = 0.169, t = 2.201, $p < 0.05$), ($\beta$ = 0.208, t = 2.037, $p < 0.05$), ($\beta$ = 0.160, t = 2.471, $p < 0.05$), and ($\beta$ = 0.376, t = 3.679, $p < 0.01$), respectively. The effect of accrual basis application controls on the quality of financial reporting was also found to be positive and significant ($\beta$ = 0.675, t = 11.461, $p < 0.01$), and therefore supported H2.

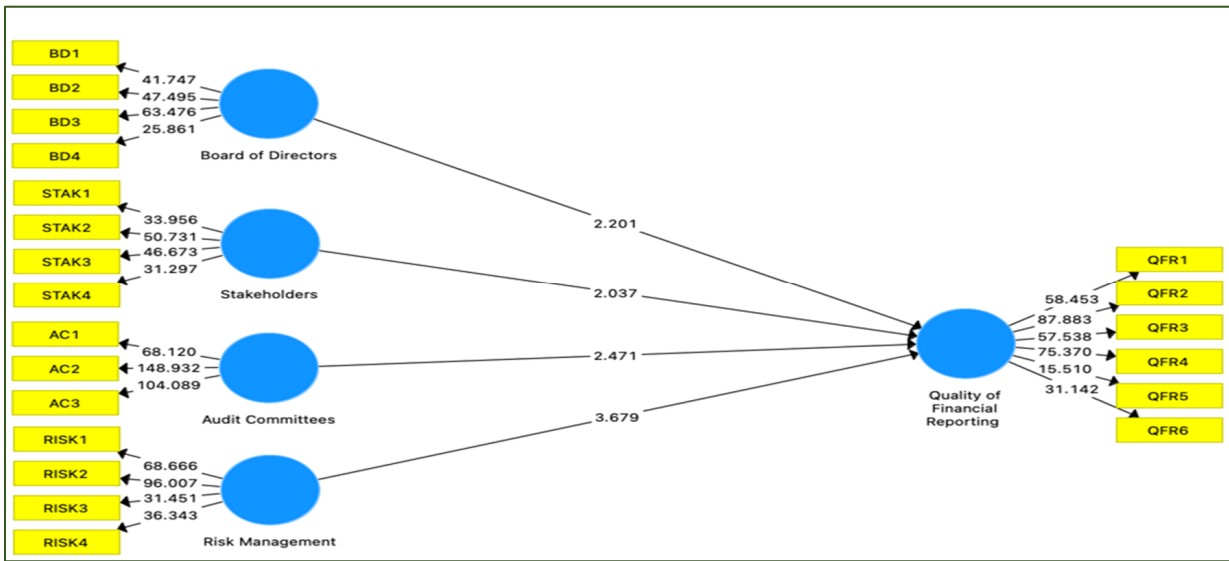

**Figure 4.** Hypotheses Testing Results (Sub-hypotheses).

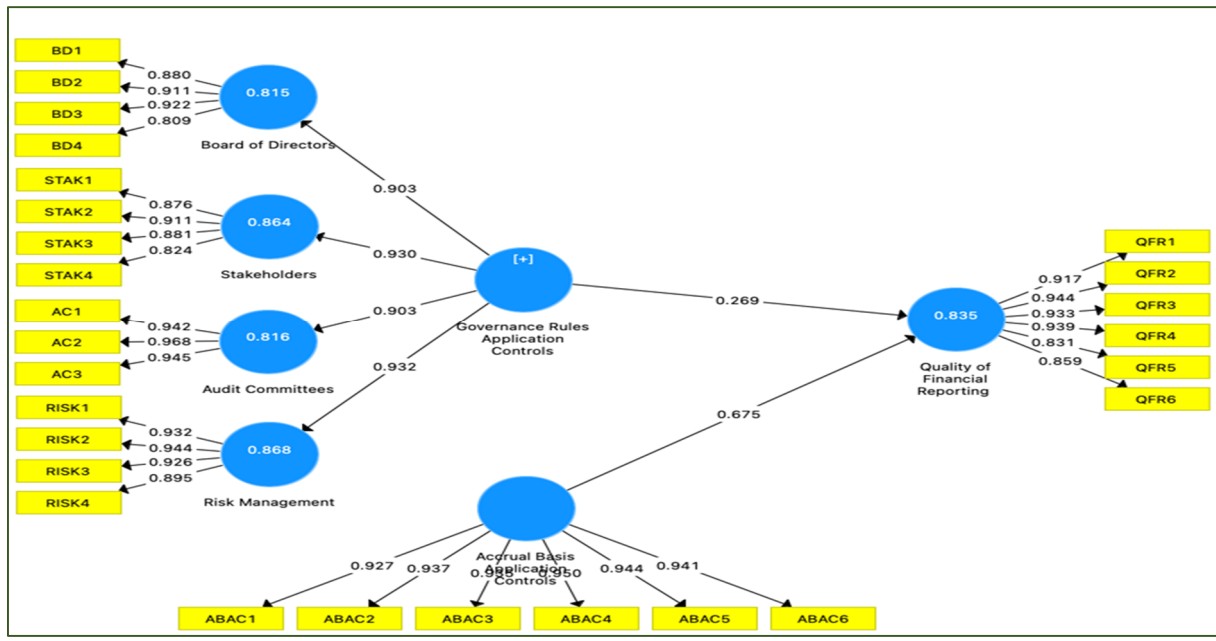

**Figure 5.** Path Coefficient.

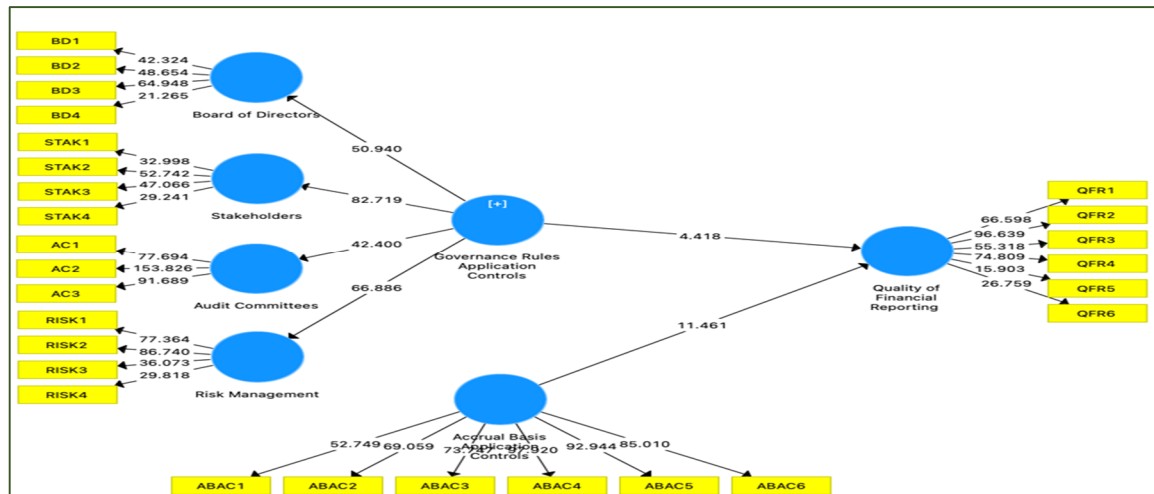

**Figure 6.** Hypotheses Testing Results.

**Table 6.** Hypotheses Testing Results.

| No | Hypothesis | Path Coefficient | Standard Error | t Value | *p* Value | Decision |
|---|---|---|---|---|---|---|
| H1 | Governance Rules Application Controls → Quality of Financial Reporting | 0.269 | 0.061 | 4.418 | 0.000 | Supported |
| H1a | Board of Directors → Quality of Financial Reporting | 0.169 | 0.077 | 2.201 | 0.028 | Supported |
| H1b | Stakeholders → Quality of Financial Reporting | 0.208 | 0.102 | 2.037 | 0.042 | Supported |
| H1c | Audit Committees → Quality of Financial Reporting | 0.160 | 0.065 | 2.471 | 0.014 | Supported |
| H1d | Risk Management → Quality of Financial Reporting | 0.376 | 0.102 | 3.679 | 0.000 | Supported |
| H2 | Accrual Basis Application Controls → Quality of Financial Reporting | 0.675 | 0.059 | 11.461 | 0.000 | Supported |

### 7.5. Predictive Relevance of the Model

The power of the function is measured using cross-validated redundancy, cross-validated, and R-square to determine the model's predictive relevance. The variation described by the dependent variable's independent (exogenous) variable is called R-square (endogenous).

Table 7 reveals that governance rules application controls and accrual basis application controls account for 84 percent of financial reporting quality. R-square is regarded large when it exceeds 0.26, moderate when it exceeds 0.13 to 0.26, and weak when it exceeds 0.02 to 0.13, according to [44]. According to these findings, these values are substantial, implying that the frameworks in this model are capable of explaining the quality of financial reporting.

**Table 7.** Prediction Relevance of the Model.

| Construct | R Square | Cross-Validity Redundancy | Cross-Validity Communality |
|---|---|---|---|
| Quality of Financial Reporting | 0.835 | 0.677 | 0.742 |

Cross-validated redundancy and cross-validated society values were used to assess the model's consistency. Their values were calculated using the SmartPLS blindfolding method. Blindfolding is a technique that involves ignoring some data values and measuring them as missing values. Following the generation of their values, a comparison will be used to determine how close the real outcomes are. The model's predictive quality must be better than 0, or the cross-redundancy values will not be verified. Table 7 displays the cross-validated redundancy results of 0.677 and the cross-validated communality results of 0.742. As a result, the value ensures that the model's predictive efficiency is appropriate.

*7.6. Goodness-of-Fit (GoF) of the Model*

According to [45]., one method is utilized to calculate the model's fitness in PLS-SEM. The average R-square and AVE geometric mean of the endogenous variable are derived in the equation below to accomplish the method:

$$\text{Gof} = \sqrt{(\overline{\text{R}^2} \times \overline{\text{AVE})}}.$$

Wetzels, Odekerken-Schröder, and Van Oppen [46] proposed the following cut-off values for GoF: (0.36 = large, 0.25 = medium, 0.1 = small). The GoF for this investigation was 0.824, which is considered large by the values in Table 8 and indicates that the model's validity is adequate.

**Table 8.** Goodness-of-Fit (GoF).

| Construct | R Square | Average Variance Extracted | Goodness of Fit |
|---|---|---|---|
| **Average** | 0.835 | **0.813** | **0.824** |

## 8. Discussion

The most crucial factor that firms seek and strive to accomplish is the quality of financial reporting. They must implement governance rules application controls and accrual basis application controls to attain the highest level of financial reporting quality. The most important drivers in improving overall financial reporting quality are governance rules application controls and accrual basis application controls, and the effects of both were investigated in this study, with mixed results. H1 is supported since the effect of governance rules application controls on financial reporting quality is positive and significant (=0.269, t = 4.418, *p* 0.001). This outcome is consistent with other research in the literature [9,26,27]. This result is related to the agency theory, which states that separating functions aids in achieving transparency and improving the organization's performance.

The study expected a link between accrual base application controls and financial reporting quality, which was empirically confirmed (=0.675, t = 11.461, *p* 0.001), confirming hypothesis H2. This finding is in line with the findings of prior research that have indicated a favorable and significant impact of accrual base application controls on financial reporting quality [4,29–31,33].

## 9. Conclusions

The purpose of this research is to see how governance regulations and accrual accounting affect the quality of financial reports in Saudi universities (Jouf University). A total of 348 questionnaires were issued to respondents, with 242 being returned. Structural equation modeling was utilized to put the postulated model to the test. The findings validated the impact of the governance rules and the accrual-based application controls on the quality of financial reporting at Jouf University.

Researchers who have a better understanding of how governance rules and the accrual foundation of application controls affect financial reporting quality will greatly benefit. This is significant because universities have long been viewed by researchers as organizations that are uninterested in strategic planning or innovation; however, innovation and strategic

planning are critical to university success. Rapid transition puts a strain on available resources and organizational systems, affecting managers' and employees' activities.

As a result of the intimate connection between the two sectors, new knowledge will be generated to fuel university activities. Furthermore, this research tried to close a gap in earlier research that looked at these linkages in the context of organizations. In reality, managers, practitioners, and decision-makers from various public and commercial companies can make use of the findings.

This research has consequences for university decision-makers in terms of how to manage organizational resources and improve financial reporting quality. Because governance rules application controls are so crucial in affecting financial reporting quality and providing a competitive advantage, establishing a governance rules application controls culture should be prioritized. The importance of the accrual basis application controls at universities must be recognized by management, as they can improve the quality of financial reporting. This indicates that, in order for such a culture to emerge, these activities must be in place. That is, the university decision-makers must understand the governance principles and the accrual foundation of application controls in order to add value to their institutions.

As a result, it is envisaged that the novel empirical findings of this study will act as a stimulus to university management, if they are taken into account. Moreover, the influence of the governance rules and the accrual basis of application controls on the quality of financial reporting at Jouf University is the focus of this study. Additionally, this research could be applied to other universities in the public and commercial sectors, investigating a model that incorporates these techniques in other firms, particularly those that are manufactured by nature to yield intriguing results.

Finally, this study has some limitations as any prior study. First, data were collected using a cross-sectional approach at a certain point in time. Longitudinal research could be used to clarify and explain the complicated relationships between the governance rules and the accrual basis of application controls on financial reporting quality over time, due to the complicated joint impact of the governance rules and the accrual basis of application controls on financial reporting quality. This method can discover changes in the relationship between the variables over time. Secondly, the chosen research methodology of the study made it difficult for the researcher to detect dynamic relationships between variables over time. Thirdly, this study focused on Jouf University. Thus, the researchers recommend future studies on other universities.

**Author Contributions:** Conceptualization, T.K.T.I.; methodology, E.M.A.-M.; software, E.M.A.-M.; validation E.M.A.-M.; formal analysis, E.M.A.-M.; investigation, E.M.A.-M. and T.K.T.I.; resources, T.K.T.I. and E.M.A.-M.; data curation, T.K.T.I.; writing—original draft preparation, T.K.T.I.; writing—review and editing, T.K.T.I. and E.M.A.-M.; visualization, E.M.A.-M.; supervision, E.M.A.-M.; project administration, T.K.T.I.; funding acquisition, T.K.T.I. All authors have read and agreed to the published version of the manuscript.

**Funding:** This research was funded by General Research Project of Scientific Research at Jouf University, grant number DSRA2020-05-522.

**Institutional Review Board Statement:** Not applicable.

**Informed Consent Statement:** Not applicable.

**Acknowledgments:** The authors would like to express their gratitude to the Deanship of Scientific Research at Jouf University for funding this work through General Research Project under grant number (DSRA2020-05-522).

**Conflicts of Interest:** The authors declare no conflict of interest.

**Appendix A**

The following is a list of total scale items used to measure the study's variables.

| Governance Rules Application Controls | |
| --- | --- |
| **Board of Directors** | |
| BD1 | The council and its main boards demonstrate a commitment to accountability and transparency. |
| BD2 | The council works to clarify the authority and responsibilities of the university president, his deans, and the deans of faculties and research centers. |
| BD3 | The council works on evaluating the performance of the higher leadership positions at the university. |
| BD4 | The board shows the results of the final account. |
| **Audit Committees** | |
| AC1 | The university has an effective control system that protects the university's assets from loss and promotes the optimal use of resources. |
| AC2 | The university has an effective system of internal control and audit that is directly linked to the council and the higher leadership of the university. |
| AC3 | There is independence in the work of the audit committees. |
| **Risk Management** | |
| RISK1 | The university is working on developing comprehensive risk management strategies. |
| RISK2 | The university works on developing, reviewing, and directing the main business plans and risk management policies. |
| RISK3 | The university has a clear mechanism for dealing with financial risks and assessing the possibility of the risk occurring. |
| RISK4 | The university has appropriate control systems to manage risks. |
| **Stakeholders** | |
| RISK1 | The university is committed to the principle of transparency and disclosure in its financial reports. |
| RISK2 | The university discloses financial and non-financial information. |
| RISK3 | The college councils at the university play an important role in providing the requirements of the labor market. |
| RISK4 | Faculty members participate in selecting administrators, preparing the budget, and defining educational policies. |
| **Accrual Basis Application Controls** | |
| ABAC1 | Recording revenue regardless of the cash collection incident. |
| ABAC2 | Keeping track of expenses regardless of the cash payment incident. |
| ABAC3 | Recording accounts receivable and payable. |
| ABAC4 | Preparing an independent financial center for the accounting unit. |
| ABAC5 | Prepare a cash flow statement to monitor period flows. |
| ABAC6 | Compliance with international accounting standards in recognition, measurement, and disclosure. |
| **Quality of Financial Reporting** | |
| QFR1 | Provide relevant information in terms of relevance and timing. |
| QFR2 | Presenting information in a simple and easy to understand manner. |
| QFR3 | Providing true, accurate, and non-distortion able information. |
| QFR4 | Adopting its information on approved professional standards. |
| QFR5 | Reviewing and approving its reports through approved regulatory bodies. |
| QFR6 | Exposing it to a system of incentives and rewards that limits manipulation. |

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
