# Peer review of "The Effect of Governance Rules Application Controls and the Accrual Basis Application Controls on Quality of Financial Reporting: Applying to Jouf University"

_sustainability, doi:10.3390/su14052831_

Round 1

Reviewer 1 Report

See appendix.

Reviewer 2 Report

First of all I have appreciated the intention of the paper. Nonetheless, there are some issues that should be fixed before publication.  1. The number of sections is too high. The introduction should be longer to include also the statement of the problem and illustrate the rules for governance and accrual basis for financial accounting. 2. The illustration of CG and accounting rules should be inserted in the literature review section, simplifying the content and summarising results from other studies (not just a list of what other authors said on the topic). 3. I'm not sure that the respondents (not homogeneous) are able to answer regarding the quality of the financial statement. 4. Questions on financial statements really convert the main aspects of QUALITY? 5. Limitation of the study and next steps not really investigated. 6. research gap could be inserted in the literature review 7. figure 1,2,3 should have a title. 8. Results are reported in figure 4 and not in figure 2 as inserted in a sentence. 9. some considerations on respondents should be inserted. 10. I would suggest separating the discussion from conclusion 11. Basically, the paper remains quite general in relation to the topic. 12. in the title I would consider being more general, 13. research gap should be in relation to all universities and not just for 1 country.  14. Try to simplify the first part and give more emphasis to the second one. 15. Use a proofreading service. 16. reduce the literature review for CG and improve the one for financial statements. Which are accounting standards? 17. first two figures really necessary? 18. A few more comments on the methods used are necessary. Good Luck. 

Reviewer 3 Report

Dear author,

I believe that the proposed manuscript can be of great value in generating knowledge and that it is a great contribution.

However, there are several points I would like to highlight:

Starting right from the abstract, the authors state as your objective to study the impact of the application of governance rules and the basis of accrual on the quality of financial reports in Saudi universities. But then your sample consisted of Jouf University staff. It seems to me that the sample is not in line with the objective. The authors should adjust the objective.

Section 5 - Quality of Financial Reporting begins by describing the purpose of the study. Here again, the authors state that "the present study aims to study the impact of the application of governance rules and the basis of accumulation on the quality of financial reporting in Saudi universities". Such an objective coincides with the one presented in the Abstract. However, the objective of the paper is not this and focuses only on the University of Jouf. So, in this case, the objective cannot be so generic and should focus only on the University of Jouf.

Before the actual literature review, the authors have several framework points, which in my opinion, could be more summarized in a single point making the article simpler.  Regarding the literature review, I believe that the authors should define an accounting theory that would support the hypothesis and possible results. Check if, for example, the institutional theory would help to support your results.

The authors state that "The researchers believe that none of these studies dealt with measuring the impact of applying the rules of governance and the accrual basis on the quality of financial reports, which is what we will discuss in the current study through the application of a sample of workers in administrative and financial units and academics who practice this at Jouf University and its impact on the quality of financial reports”. Here I only question whether this is the opinion of other researchers, in which case you should indicate which ones, or whether it is your opinion after the literature review that has been conducted.

In the methodology, the authors mention how the sample is made up, being broken down among University staff (academic, administrative, and financial), but they do not mention the total population broken down by these three types of employees. That is, I ask if the sample is representative, because nothing was mentioned regarding the representativeness of the sample.

In the presentation of the results, the authors, should not give the theoretical framework of the model (point 8.3). At most this could be in the methodology, but I don't think it is necessary.

In formal terms, since the article is spread over many points and sub-points, then they create some confusion. Take, for example, the sub-point (8.1.1) is repeated. Also, all the sub-points in (8.3) are incorrect and should be changed.

In the text, the authors do not refer to table 5. I think that when they refer (line 690) to table 3 it should be table 4 and then when they refer to table 4 (line 704), it should be table 5.

In the conclusions, the authors should make reference to whether they are only for the university they are studying, whether they are conclusions only for the respondents, or whether they can generalize them. On the other hand, they identify the gaps but do not define leads for future research.

Round 2

Reviewer 2 Report

Section 2 could be inserted in a wider introduction.  section 3 is a bit redundant for the purpose of the paper. Globally speaking the paper must better explain why there is a supposed relationship between input and output variables, what is the meaning and why results are important and have an impact. In other words, simplify and restructure the paper.

Author Response

Dear Editor of sustainability journal

 We would like to thank you and reviewers for your great comments that totally improve the paper. We would like to inform you that we taken all comments and modified as you per requested. Finally, if there are any comments, it is our pleasure to make it fast.

Your cooperation is highly appreciated

Reviewer :2:

Section 2 could be inserted in a wider introduction. 

Done

Please refer to page 2-3

section 3 is a bit redundant for the purpose of the paper.

Done

We have reduced as you per requested. Please refer page 4-5

Globally speaking the paper must better explain why there is a supposed relationship between input and output variables, what is the meaning and why results are important and have an impact. In other words, simplify and restructure the paper.

Done. Thank you so much for this comments that totally will improve the paper.

We have rechecked some part and already reduced and added some important point as you per requested. Please refer to 3-5, 7 and 22.

Reviewer 3 Report

I think that the article is now better than in the previous version. However, it can still be improved.

I think the purpose could be even clearer, that is, "The purpose of this research is to see how governance regulations and accrual accounting affect the quality of financial reports in Jouf University (Saudi university)"

The authors have narrowed down the framework they had before the literature review. At this point, I still think it could be further simplified. It has many points and sub-points, each with one paragraph, which is not justified. Do you really think it is necessary to have so many points and sub-points? Wouldn't the article be more fluid if the text ran without so many headings? In my opinion, you could summarize this framework in a single point, making the article simpler. 

Regarding the literature review, I believe that the authors should define an accounting theory that supports the hypotheses and possible results. Check if for example institutional theory would not help to support your results. It would help explain the results and support the conclusions.

Author Response

Dear Editor of sustainability journal

 We would like to thank you and reviewers for your great comments that totally improve the paper. We would like to inform you that we taken all comments and modified as you per requested. Finally, if there are any comments, it is our pleasure to make it fast.

Your cooperation is highly appreciated

Reviewer :3:

I think that the article is now better than in the previous version. However, it can still be improved.

Thank you so much for your support.

I think the purpose could be even clearer, that is, "The purpose of this research is to see how governance regulations and accrual accounting affect the quality of financial reports in Jouf University (Saudi university)"

Done

Please refer to page 1

The authors have narrowed down the framework they had before the literature review. At this point, I still think it could be further simplified. It has many points and sub-points, each with one paragraph, which is not justified. Do you really think it is necessary to have so many points and sub-points? Wouldn't the article be more fluid if the text ran without so many headings? In my opinion, you could summarize this framework in a single point, making the article simpler.

Done

Regarding the literature review, I believe that the authors should define an accounting theory that supports the hypotheses and possible results. Check if for example institutional theory would not help to support your results. It would help explain the results and support the conclusions

Done. Thank you so much for this comments that totally will improve the paper.

We have rechecked some part and already reduced and added some important point as you per requested. Please refer to 3-5 and 22.

Round 3

Reviewer 2 Report

Dear Authors, the paper reaches the minimum level for being published after 3 reviews but its contribution to the literature remains limited due to its structure (redundant) and because some assumptions are not motivated. Good luck for the next steps.